# Thrombocytopenia and Intracranial Venous Sinus Thrombosis after “COVID-19 Vaccine AstraZeneca” Exposure

**DOI:** 10.3390/jcm10081599

**Published:** 2021-04-09

**Authors:** Marc E. Wolf, Beate Luz, Ludwig Niehaus, Pervinder Bhogal, Hansjörg Bäzner, Hans Henkes

**Affiliations:** 1Neurologische Klinik, Klinikum Stuttgart, D-70174 Stuttgart, Germany; ma.wolf@klinikum-stuttgart.de (M.E.W.); h.baezner@klinikum-stuttgart.de (H.B.); 2Department of Neurology, Universitätsmedizin Mannheim, University of Heidelberg, D-68167 Mannheim, Germany; 3Zentralinstitut für Transfusionsmedizin und Blutspendedienst, Klinikum Stuttgart, D-70174 Stuttgart, Germany; b.luz@klinikum-stuttgart.de; 4Neurologie, Rems-Murr-Klinikum Winnenden, D-71364 Winnenden, Germany; ludwig.niehaus@rems-murr-kliniken.de; 5Department of Interventional Neuroradiology, The Royal London Hospital, Barts NHS Trust, London E1 1FR, UK; bhogalweb@gmail.com; 6Neuroradiologische Klinik, Klinikum Stuttgart, D-70174 Stuttgart, Germany; 7Medical Faculty, University Duisburg-Essen, D-47057 Duisburg, Germany

**Keywords:** COVID-19 vaccine AstraZeneca, venous sinus thrombosis, platelet factor 4 antibodies, rheolysis, anticoagulation

## Abstract

Background: As of 8 April 2021, a total of 2.9 million people have died with or from the coronavirus infection causing COVID-19 (Corona Virus Disease 2019). On 29 January 2021, the European Medicines Agency (EMA) approved a COVID-19 vaccine developed by Oxford University and AstraZeneca (AZD1222, ChAdOx1 nCoV-19, COVID-19 vaccine AstraZeneca, Vaxzevria, Covishield). While the vaccine prevents severe course of and death from COVID-19, the observation of pulmonary, abdominal, and intracranial venous thromboembolic events has raised concerns. Objective: To describe the clinical manifestations and the concerning management of patients with cranial venous sinus thrombosis following first exposure to the “COVID-19 vaccine AstraZeneca”. Methods: Patient files, laboratory findings, and diagnostic imaging results, and endovascular interventions of three concerning patients were evaluated in retrospect. Results: Three women with intracranial venous sinus thrombosis after their first vaccination with “COVID-19 vaccine AstraZeneca” were encountered. Patient #1 was 22 years old and developed headaches four days after the vaccination. On day 7, she experienced a generalized epileptic seizure. Patient #2 was 46 years old. She presented with severe headaches, hemianopia to the right, and mild aphasia 13 days after the vaccination. MRI showed a left occipital intracerebral hemorrhage. Patient #3 was 36 years old and presented 17 days after the vaccination with acute somnolence and right-hand hemiparesis. The three patients were diagnosed with extensive venous sinus thrombosis. They were managed by heparinization and endovascular recanalization of their venous sinuses. They shared similar findings: elevated levels of D-dimers, platelet factor 4 antiplatelet antibodies, corona spike protein antibodies, combined with thrombocytopenia. Under treatment with low-molecular-weight heparin, platelet counts normalized within several days. Conclusion: Early observations insinuate that the exposure to the “COVID-19 vaccine AstraZeneca” might trigger the expression of antiplatelet antibodies, resulting in a condition with thrombocytopenia and venous thrombotic events (e.g., intracranial venous sinus thrombosis). These patients’ treatment should address the thrombo-embolic manifestations, the coagulation disorder, and the underlying immunological phenomena.

## 1. Introduction

Intracranial venous sinus thrombosis (IVST) is a potentially life-threatening condition with an incidence of about 15 per million per year in western countries [1,2]. It is associated with various types of thrombophilia, systemic diseases, or local causes such as trauma and infection [3].

Thrombotic thrombocytopenic purpura (TTP) is observed with an annual incidence of 1 per 1,000,000 and is a potential sequela, among others, of virus infection and vaccination [4,5].

A differential diagnosis in iatrogenic auto-immune-mediated thrombocytopenia is heparin-induced thrombocytopenia (HIT), with a frequency of approximately 0.2% in hospitalized patients receiving thromboprophylaxis [6].

In early March 2021, a total of nine patients with thrombocytopenia and IVST after vaccination with “COVID-19 vaccine AstraZeneca” were reported in Germany (Paul Ehrlich Institute, communication on 19 March 2021). Two of these patients and a third one a few days later were identified and treated by the authors. Herein, the clinical courses and the patients’ key findings are summarized, especially regarding treatment options such as endovascular thrombectomy as acute rescue therapy. Furthermore, similarities of the patients might help one to understand the underlying pathophysiology better. The patients underwent medicinal and endovascular treatment. Thus far, the clinical course was relatively benign in our three patients, while at least two patients with the same condition treated elsewhere passed away.

## 2. Methods

The clinical, laboratory, and imaging findings and the results of endovascular and medicinal interventions were analyzed in retrospect. Attempts were undertaken to both find similarities and differences between the three cases. Based on this data, preliminary conclusions were drawn concerning the cause and treatment of this severe condition.

The following laboratory tests had been used:
Anti-factor Xa (anti-Xa) assay: COAMATIC HEPARIN (Chromogenix, Bedford, MA USA) calibrated for low-molecular-weight heparin (LMWH) or rivaroxaban or apixaban, a calibration for danaparoid is not available. The anti-factor Xa-activity calibrated against LMWH was used, which might yield false-low results.Antithrombin activity: Berichrom Antithrombin III (A) (Siemens, Munich, Germany);ARCHITECT SARS-CoV-2 IgG (Abbott, Abbott Park, IL USA)Beta-2-glycoprotein I antibody test: ELISA (Euroimmun, Lübeck, Germany);Cardiolipin antibody: ELISA (Euroimmun)Hemostasis analyzer: CS5100-Gerät (Sysmex, Kobe, Japan)Heparin-induced platelet activation (HIPA), according to Greinacher et al. (https://www2.medizin.unigreifswald.de/transfus/fileadmin/user_upload/doku_thrombo_gerinnung/anleitung_hipa.pdf; accessed on 1 March 2021)Lupus anticoagulant (functional): Pathromtin SL (aPTT) (Siemens) and LA 1 Screen (dRVVT)/LA2 Confirm (dRVVT) (Siemens)Heparin-induced thrombocytopenia platelet factor 4 antibody (PF4)-IgG EIA (Immucor, Dreieich, Germany) (in the presence of high heparin concentration the ELISA was negative)Protein C activity: ProC AcR (Siemens)Protein S activity: Protein S Ac (Siemens)SARS-CoV-2 antibody test: LIAISON SARS-CoV-2 S1/S2 IgG (DiaSorin, Saluggia, Italy).

## 3. Results

Three women with thrombocytopenia and IVST following a first “COVID-19 vaccine AstraZeneca” exposure were so far encountered, and their clinical courses and treatments are described.

### 3.1. Patient #1

A 22-year-old woman had no significant previous medical history and denied smoking and oral contraception. Her body mass index upon admission was 26. Following a first AstraZeneca vaccination, she observed shivering, fever and headaches for two days, with spontaneous resolution. On day 4 after the vaccination, she developed new frontally accentuated headaches, and on day 7, a self-limited generalized epileptic seizure occurred. The pronator drift test was positive for her left arm. MRI showed blood in the subarachnoid space adjacent to the falx cerebri on both sides. The superior sagittal sinus, the left-hand transverse sinus, and the sigmoid sinus were thrombosed. Laboratory examinations revealed thrombocytopenia with a platelet count of 75,000/μL. The SARS-CoV-2-XPCR was negative; the SARS-CoV-2 spike-antigen antibody level was 19.8 AU/mL. The test for heparin-induced thrombocytopenia (HIT) with clivarin, heparin sodium, enoxaparin, and danaparoid was negative. The enzyme-linked immunosorbent assay (ELISA) for IgG antibodies against the platelet factor 4-heparin complex was positive. Extensive tests for thrombophilia (Factor V R5066 (Leiden) mutation, prothrombin mutation, antiphospholipid antibodies, antinuclear antibodies, protein C activity, protein S activity) were negative or inconspicuous.

Digital subtraction angiography (DSA) confirmed the occlusion of the ascending cerebral veins and the said sinuses. Endovascular rheolysis (i.e., flushing with an aspiration of a 50% saline solution and contrast medium mixture inside the concerning sinuses via a large-caliber catheter (SOFIA Plus, MicroVention, Aliso Viejo, CA USA) resulted in a complete recanalization of the sinuses (Figure 1).

The patient received 2 × 1000 mg levetiracetam (Keppra, UCB Pharma, Monheim, Germany) per os (PO) daily (initial dose IV) for three months and 2 × 80 mg enoxaparin sodium (Clexane, Sanofi-Aventis, Paris, France) subcutaneous (SC) daily for ten days, followed by direct oral anticoagulation with 2 × 150 mg dabigatran (Pradaxa, Boehringer-Ingelheim, Ingelheim, Germany) PO daily for six months.

Follow-up MRI on day 5 and day 10 after the endovascular procedure confirmed the intracranial venous sinuses’ recanalization with neither new hemorrhage nor edema.

When the patient was transferred to a rehabilitation facility 20 days after the vaccination and 13 days after the epileptic seizure, her clinical status was rated 0 according to the modified Rankin scale (mRS). Follow-up laboratory examinations showed a spontaneous increase in the platelet count to 97,000/μL on early follow-up and complete normalization was confirmed after two weeks.

### 3.2. Patient #2

A 46-year-old woman without previous disease, non-smoking, and not using oral contraception presented severe headaches eight days after her first AstraZeneca vaccination. Five days later, she developed focal neurologic symptoms with mild aphasia and hemianopia to the right (13 days after vaccination). The vaccination was initially well tolerated. Upon admission, she was somnolent.

MRI showed a thrombotic occlusion of the superior sagittal sinus and the left-hand transverse sinus and sigmoid sinus. An acute intracerebral hematoma with a diameter of 30 mm was seen in the left occipital lobe.

Laboratory examinations showed thrombocytopenia with a platelet count of 60,000/µL. The SARS-CoV-2-XPCR was negative, the SARS-CoV-2 S1/S2-IgG antibody level was 19.1 AU/mL, confirming the vaccine-induced anti-S protein antibody formation. The test for HIT with clivarin, heparin sodium, enoxaparin, and danaparoid was negative. An IgG-ELISA for platelet factor 4-heparin complex antibodies was positive. Extensive tests for thrombophilia markers (protein S activity, protein C activity, lupus inhibitors, homocysteine) were negative.

DSA demonstrated the occlusion of the superior sagittal sinus and the left transverse sinus and the sigmoid sinus. Partial thrombus removal was achieved by endovascular rheolysis in two separate sessions, including balloon angioplasty of the transition from the left internal jugular vein to the sigmoid sinus.

The patient received 2 × 80 mg enoxaparin sodium SC daily for two days. Since a HIT was suspected, the medication was changed to 3 × 750 mg danaparoid (Orgaran, Aspen Pharmacare, Durban, South Africa) SC daily, which did not achieve a sufficient factor Xa inhibition, as measured by a non-calibrated anti-factor Xa assay. The same was true for the increased dosage of 4 × 1250 mg danaparoid SC daily. The non-calibrated anti-factor Xa assay may yield false-low results. Direct oral anticoagulation with 2 × 150 mg dabigatran PO daily for six months was initiated on day 14 after the clinical onset.

Follow-up MRIs on days 3, 5, and 12 after the endovascular procedure confirmed the complete recanalization of the superior sagittal sinus and partial recanalization of the left transverse sinus and sigmoid sinus, without new hemorrhage nor progressive edema.

The patient was transferred to a rehabilitation facility on day 27 after the vaccination and on day 14 after the endovascular treatment. Her clinical status was rated mRS 1. Follow-up laboratory examinations showed a spontaneous increase in the platelet count to 228,000/mL on day 11 after the clinical onset.

### 3.3. Patient #3

A 36-year-old female patient, lean and without previous health issues, non-smoking and not using oral contraception, presented with severe headaches seven days after a first AstraZeneca vaccination, followed by three days of fever and headache. Since the headache did not respond to conventional analgesics, she was referred to an ear, nose, and throat (ENT) physician with suspected sinusitis, but the examination was unrevealing. After ten days of headache (day 17 after vaccination), she developed acute somnolence with a right-hand hemiparesis.

MRI showed a thrombotic occlusion of the straight sinus and a non-occlusive thrombus in the superior sagittal sinus. Congestive edema of both thalami was more pronounced on the left side. The patient received 2250 IU danaparoid SC and was transferred to us.

Laboratory examinations showed thrombocytopenia with a platelet count of 92,000/µL. The SARS-CoV-2-XPCR was negative; the SARS-CoV-2 spike-antigen antibody level was 75 AU/mL. The test for HIT with clivarin, heparin sodium, enoxaparin, and danaparoid was negative. An ELISA for IgG antibodies against the platelet factor 4-heparin complex was positive. Standard tests for thrombophilia were negative.

DSA demonstrated the occlusion of the straight sinus and a non-occlusive thrombus of the superior sagittal sinus. Significant thrombus removal was achieved by endovascular rheolysis.

The patient received 2 × 60 mg enoxaparin sodium SC daily for one week, followed by direct oral anticoagulation with 2 × 150 mg dabigatran PO daily for six months.

Follow-up MRIs on days 1 and 3 after the endovascular procedure confirmed the intracranial venous sinuses’ recanalization and neither new hemorrhage nor progressive edema.

Upon submitting the manuscript, the asymptomatic (mRS 0) patient was already transferred to a rehabilitation facility. Follow-up laboratory examinations showed a spontaneous increase in the platelet count to 150,000/µL on day 3 after the clinical onset.

### 3.4. Common Features of the Three Patients

Three women of childbearing age without a significant medical history received a first “COVID-19 vaccine AstraZeneca”. Within 7, 13, and 17 days, they presented with symptomatic IVST and mild to moderate thrombocytopenia (60.000–92.000/µL). IgG against PF4-heparin complex was found. SARS-CoV-2-XPCRs were negative, and SARS-CoV-2 spike-protein antibody levels were elevated. Thrombophilia markers were negative. All patients received low-molecular-weight heparin (LMWH). They showed a continuous improvement of the platelet count and thrombocytopenia spontaneously resolved within <2 weeks during the following course without immunoglobulin administration or plasmapheresis. The key findings are summarized in Table 1.

## 4. Discussion

### 4.1. Pathophysiology

#### 4.1.1. Intracranial Venous Sinus Thrombosis (IVST)

The thrombotic occlusion of intracranial venous sinuses is a rare event. Reported numbers vary between 3–40 cases per million per year [7]. About 12% of all IVSTs are considered “idiopathic” after the exclusion of acknowledged risk factors. Predisposing variants of thrombophilia include antithrombin, protein C and protein S deficiency, prothrombin polymorphism, factor V Leiden mutation, antiphospholipid antibodies, hyper-homocysteinemia, pregnancy, puerperium, surgery, cancer, immobilization, oral contraceptives, and inflammatory bowel disease. Any combination of such factors may increase the risk of IVST [8,9].

#### 4.1.2. IVST and COVID-19

Since the initial outbreak of SARS-CoV-2 infections in December 2019, various mechanisms of neurological manifestations of this disease have become apparent [10]. IVST is among the less frequent but well-documented sequelae of COVID-19 [11]. Gunduz (2021) [12] reported the case of a pregnant woman with COVID-19-associated IVST with antinuclear antibodies and heterozygous prothrombin mutation, indicating thrombophilia as a predisposing factor. Thrombophilia screening yielded positive results in several other patients [13], including the occurrence of transient antiphospholipid antibodies [14].

The most frequently reported treatment was anticoagulation with heparin. Endovascular recanalization of the thrombosed sinus was infrequently carried out [15]. The mortality of COVID-19-associated IVST is in the range of 35–50% [16,17].

#### 4.1.3. Thrombocytopenia and Thromboembolic Complications

Thromboembolic complications despite thrombocytopenia (blood platelet count < 100,000/mL) appears to be a common feature of the patients with IVST after “COVID-19 vaccine AstraZeneca” immunization.

“COVID-19 vaccine AstraZeneca” uses as a vector the modified, non-replicant chimpanzee adenovirus ChAdOx1. Thrombocytopenia has been reported in the context of adenovirus replication [18].

Thrombocytopenia or IVST has also been reported for several other vaccinations:Mittelmeier (1959) [19]: active diphtheria vaccination in a 15-month-old boy who died two days after the second vaccination; autopsy showed generalized endothelial damage and thrombosis of the superior sagittal and straight sinus;Nieminen et al. (1993) [20]: measles–mumps–rubella vaccination, incidence 23/700,000, benign course, no thromboembolic complications;Hamiel et al. (2016) [21]: pediatric case, three occurrences;Yamamoto et al. (2020) [22]: influenza vaccination, the onset of alveolar hemorrhage within one day after exposure.

A comprehensive summary of the subject of vaccination-associated immune thrombocytopenic purpura (ITP), including antiplatelet antibody-negative cases, has been given by Perricone et al. (2014) [23].

While hemorrhagic complications of ITP are usually at the forefront, venous thrombotic events (VTEs) are possible. Tana et al. (2021) [24] performed a meta-analysis of the published literature on primary ITP and found an increased incidence of cerebral venous thrombosis in younger women. They emphasize that low platelet counts do not protect ITP patients from VTEs. Anticoagulation for patients with cerebral venous thrombosis and very low platelet count is associated with a high risk of fatal outcomes. Anticoagulation should be initiated with a platelet count >30,000–50,000/µL.

An immune-mediated thrombotic thrombocytopenic purpura (iTTP) was discussed [25], however, the ADAMTS13 (a disintegrin and metalloprotease with thrombospondin type 1 motifs, member 13) enzyme activity was not significantly reduced in any of our three patients.

In all three patients, antibodies against platelet factor 4 (CXCL4) were found. These antiplatelet factor 4 antibodies form a complex with the CXCL4 platelet factor and bind to the thrombocyte membrane’s Fcγ-receptor IIa. The consumption of platelets, thrombocytopenia, and disseminated intravascular coagulation ensue. This pathomechanism has similarities with heparin-induced thrombocytopenia (HIT) [26]. COVID-19 has been described as a possible predisposing factor for HIT [27].

In the three patients treated by us, thrombocytopenia after the “COVID-19 vaccine AstraZeneca” was a *transient* phenomenon and resolved short-term without specific measures.

### 4.2. Therapeutic Considerations

The management of IVST depends on the extent of the vessel occlusion and the sequelae of the venous drainage impairment. The mainstay of IVST management is full-dose heparinization [28,29]. The threshold in patients with thrombocytopenia is in the range of 50,000 platelets/µL. Endovascular recanalization of IVST is an add-on option for patients with extensive sinus thrombosis and is limited to the sinus lumen. The controversy over the technical details of how to perform this type of procedure is ongoing. In our experience, rheolysis and aspiration thrombectomy are both safe and efficacious, although not necessarily straightforward [30]. Partial thrombectomy may be sufficient to allow for further recanalization with continued heparinization [31]. As for many endovascular stroke procedures, the key is the proper timing (i.e., early treatment). Once the vicious circle of brain swelling and impaired drainage has started, decompressive craniectomy remains the last resort [32].

In COVID-19 patients treated with unfractionated heparin, the formation of antibodies against the PF4-heparin complex has been reported. This was not the case in patients treated with low molecular weight heparin [33].

Our patients #1 and #3 were treated with low molecular weight heparin (enoxaparin sodium) in therapeutic dosage during the acute phase. Patient #2 received, after two days with enoxaparin sodium, 3 × 750 mg danaparoid sodium SC daily. The anti-factor Xa assay revealed an insufficient factor Xa inhibition, which was also the case after an increase in the dosage to 4 × 1250 mg danaparoid sodium SC daily.

After one week, the heparin administration was stopped, and the three patients were further anticoagulated with 2 × 150 mg dabigatran PO daily for six months [34].

For the treatment of thrombotic thrombocytopenic purpura (TTP), glucocorticoids, intravenous immunoglobulin, and plasmapheresis are standard of care. This is also true for COVID-19 patients with this hematological disorder [35,36]. In our three patients with thrombocytopenia and IVST after “COVID-19 vaccine AstraZeneca” exposure, so far, no specific treatment of the thrombocytopenia was required.

The observation of the three patients who are described in this paper cannot be taken as evidence for the causal relation between “COVID-19 vaccine AstraZeneca” exposure and venous thrombosis, although similar incidents have been reported from several European countries. None of our three patients had any disorder or medication that could have explained the occurrence of an IVST. The majority of reported cases concerned women, and this female preponderance remains unexplained so far.

## 5. Conclusions

Thrombocytopenia and IVST might be a rare sequel of “COVID-19 vaccine AstraZeneca” exposure. The trigger is unknown so far. The pathomechanism is presumably the formation of antibodies against PF4, causing platelet consumption with low platelet counts and thrombus formation. Predisposing factors were not yet identified. Patients with reemerging severe headaches or other neurological signs and symptoms apart from a benign immunoreaction for about two days after “COVID-19 vaccine AstraZeneca” immunization should undergo laboratory and clinical examinations. Thrombocytopenia with headaches or a neurological deficit in this setting should trigger cranial MRI/MRA examinations. Patients with confirmed IVST must be hospitalized in a dedicated neurovascular center. Heparinization or anticoagulation with close follow-up of the platelet count is crucial for the treatment. Extensive venous sinus occlusion, associated with intracranial hemorrhage, epileptic seizures, focal neurological deficit, and impaired consciousness, requires aggressive and escalating interventions. Endovascular recanalization of thrombosed venous sinuses using rheolysis is the first step. Decompressive craniectomy is required in the case of massive brain swelling. Early diagnosis and compelling treatment of IVST and the underlying coagulation disorder is the only way to prevent handicap or death of these patients. Pooling the concerned patients’ data might help improve our understanding of this condition and identify patients at risk upfront, hopefully.

The risk—benefit consideration of using different COVID-19 vaccines is beyond the scope of this paper.

## Figures and Tables

**Figure 1 jcm-10-01599-f001:**
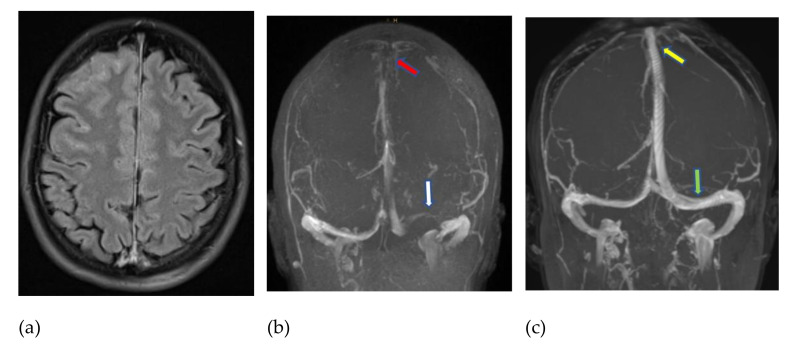
MRI (fluid-attenuated inversion recovery—FLAIR) obtained 7 days after a first AstraZeneca vaccination and immediately after a generalized epileptic seizure, showing blood in the subarachnoid space (**a**). Venous time-of-flight MR angiography (**b**) revealed thrombotic occlusion of the superior sagittal sinus (red arrow) and left transverse sinus (white arrow). An early follow-up examination ten days later (**c**), under full-dose heparinization supported by endovascular thrombectomy of the sinuses, confirmed the recanalization of the superior sagittal sinus (yellow arrow) and left-hand transverse sinus (green arrow).

**Table 1 jcm-10-01599-t001:** Summarized clinical characteristics and laboratory findings in 3 patients with intracranial venous sinus thrombosis after “COVID-19 vaccine AstraZeneca” exposure.

	Patient #1	Patient #2	Patient #3
Age (years)	22	46	36
Onset of headache(days after vaccination)	4	8	7
Focal neurologic symptoms (days after vaccination)	7	13	17
Focal symptoms	generalized epileptic seizures	mild aphasia, homonymous hemianopia to the right	aphasia, reduced consciousness
Platelet count upon admission (n/µL)	75.000	60.000	92.000
COVID-spike-Ab (AU/mL)	+(19.8)	+(19.1)	+(75)
HIPA-Ab	−	−	−
PF4-Ab	+	+	+
ADAMTS-13-activity	normal (76%)	normal (73%)	normal (86%)
Antiphopholipid-Abs	−	−	−
D-Dimer (ng/mL)	2590	22,800	2120
Protein C activity	normal	normal	normal
Protein S activity	normal	normal	normal
Prothrombin G20210A	wild type	wild type	wild type
Factor V R5066 (Leiden)	wild type	wild type	wild type
Homocysteine (µmol/L)	19.7	5.5	17.1
MTHFR	wild type	NA	heterozygote mutation (C667T)

HIPA = heparin-induced platelet activation; Ab = antibody; PF4 = platelet factor 4; ADAMTS = a disintegrin and metalloproteinase with thrombospondin motifs enzyme; MTHFR = methylene tetrahydrofolate reductase.

## Data Availability

The data presented in this study is available on request from the corresponding author. The data is not publicly available due to patient’s privacy protection.

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
