# Peer review of "Thrombocytopenia and Intracranial Venous Sinus Thrombosis after “COVID-19 Vaccine AstraZeneca” Exposure"

_jcm, 2021, doi:10.3390/jcm10081599_

Round 1

Reviewer 1 Report

This article is a well organized discussion of the current thrombotic Adverse Reactions of COVID-19 AstraZeneca Vaccine. The topic wasn't super technique, but brought in multiple sources on the current AstraZeneca vaccine.

This discovery should be viewed with caution. I'm not sure that the vaccine is to blame for the blood clots that have been connected to it (though I do think that vaccination cannot be excluded as a cause based on the information available). Furthermore, these clots are exceedingly rare, and the vaccine's advantages in terms of preventing serious Covid-19, avoiding hospitalization, and avoiding death greatly outweigh the clots' possible risks.

  • A major limitation is that the study sample is not fully representative of the general population, as it represents a self‐selected group of individuals (women). The study sample is little and also composed predominantly of women  compared with those observed in hospital settings.
  • Secondly, some of these manifestations could have been caused by adverse reactions to drugs used to treat SARS‐CoV‐2 and/or for other purposes. Are there any pre‐existing diseases of the patients under investigation to consider if they influenced the rate of the reported events?
  • I recommend including a list of all abbreviations used in the text and paying attention to write the full names of the acronyms reported in the text.
  • I recommend including also a table with all the patient specifications.

Sincerely.

Author Response

Reviewer #1

The authors are grateful for the proposals of reviewer #1.

The text has undergone language revision by a native US English speaker with a medical background (AC).

<< This discovery should be viewed with caution. I'm not sure that the vaccine is to blame for the blood clots that have been connected to it (though I do think that vaccination cannot be excluded as a cause based on the information available).

>> The following has been added to the text: The observation of the three patients who are described in this paper cannot be taken as evidence for the causal relation between “COVID-19 Vaccine AstraZeneca” exposure and venous thrombosis, although similar incidents have been reported from several European countries. None of our three patients had any disorder or medication that could have explained the occurrence of an IVST.

<< Furthermore, these clots are exceedingly rare, and the vaccine's advantages in terms of preventing serious Covid-19, avoiding hospitalization, and avoiding death greatly outweigh the clots' possible risks.

>> This is an aspect beyond the scope of our manuscript. The following has been added to the text: The risk-benefit consideration of using different COVID-19 vaccines is beyond the scope of this paper.

<< A major limitation is that the study sample is not fully representative of the general population, as it represents a self‐selected group of individuals (women). The study sample is little and also composed predominantly of women compared with those observed in hospital settings.

>> It was not our goal to undertake an investigation into the safety profile of the “COVID-19 Vaccine AstraZeneca”. We report the clinical cases of three women because apparently mainly women are affected by the post-vaccination thrombo-embolic events. The following has been added to the text: The majority of reported cases concerned women, and this female preponderance remains unexplained so far.

<< I recommend including a list of all abbreviations used in the text and paying attention to write the full names of the acronyms reported in the text.

>> The following list of abbreviations has been added: Abbreviations: ADAMTS13: a disintegrin and metalloprotease with thrombospondin type 1 motifs, member 13; COVID-19: coronavirus disease 2019; DSA: digital subtraction angiography; ELISA: enzyme-linked immunosorbent assay; ENT: ear, nose, and throat physician; FLAIR: fluid-attenuated inversion recovery; HIPA: Heparin-induced platelets-aggregate; HIT: heparin-induced thrombocytopenia; ITP: immune thrombocytopenic purpura; iTTP: immune-mediated thrombotic thrombocytopenic purpura; IU: international units; IV: intravenous; IVST: intracranial venous sinus thrombosis; LMWH: low-molecular-weight heparin; MRA: magnetic resonance angiography; MRI: magnetic resonance imaging; mRS: modified Rankin scale; PF4: Heparin-induced Thrombocytopenia Platelet Factor 4 Antibody; PO: per os; SARS: severe acute respiratory syndrome; SC: subcutaneous; TTP: thrombotic thrombocytopenic purpura; VTE: venous thrombotic events

<< I recommend including also a table with all the patient specifications.

>> Table 1 has been added:

Table 1: Summarized clinical characteristics and laboratory findings in 3 patients with intracranial venous sinus thrombosis after “Covid-19 Vaccine AstraZeneca” exposure.

Patient #1

Patient #2

Patient #3

Age [years]

22

46

36

Onset of Headache [days after vaccination]

4

8

7

Focal neurologic symptoms

[days after vaccination]

7

13

17

Focal symptoms

generalized epileptic seizures

mild aphasia, homonymous hemianopia to the right

aphasia, reduced consciousness

Platelet count on admission [n/µl]

60.000

75.000

92.000

COVID-spike-Ab [AU/ml]

+ (19.8)

+ (19.1)

+ (75)

HIPA-Ab

-

-

-

PFA-4-Ab

+

+

+

ADAMTS-13-Level

normal

normal

normal

Antiphopholipid-Ab

-

-

-

D-Dimer [ng/ml]

2590

8780

2120

Protein C activity

normal

normal

normal

Protein S activity

normal

normal

normal

Prothrombin G20210A

wild type

wild type

wild type

Factor V R5066
(Leiden)

wild type

wild type

wild type

Homocysteine [µmol/l]

19.7

5.5

17.1

MTHFR

wild type

NA

pending

HIPA=Heparine-induced platelet activation; Ab=antibody; PFA=Platelet factor activation; ADAMTS=A Disintegrin And Metalloproteinase with ThromboSpondin motifs enzyme; MTHFR=Methylene tetrahydrofolate reductase

Reviewer 2 Report

Wolf et al describe three cases of intracranial venous sinus thrombosis after exposure to the COVID-19 vaccine from AstraZeneca. The cases are well described and the information given by the authors is of potentially high value for physicians treating such patients. I would like to urge the editors to make that relevant contribution available to the public as soon as possible. I have only a few comments to make:

Major

  1. The authors should be very specific with the tests that were applied! Most of the relevant tests are very specific to the manufacturer and cannot be compared between laboratories. Which PF4-ELISA was used (manufacturer & modifications, if any); which SARS-CoV-2 antibody test (manufacturer & modifications, if any) was used; which antiphospholipid antibody testing (which antibodies where tested, beta2-gylcoprotein I and cardiolipin both as IgG and IgM I assume, in which assay), Did the authors also test for a functional lupus anticoagulant? In addition, please make clear how protein S was assessed (free protein S would be the recommended test, but that doesn’t give you “protein S acitivity”), same for protein C (chromogenic assay would be the recommended test). Please state if patients were tested for antithrombin activity. It was maybe low, which can be expected in the clinical setting. Apparently some type of HIPA was performed. Please cite a reference for this test.
  2. The authors explain that with danaparoid, they did not reach sufficient factor Xa inhibition. I assume that danaparoid was given SC and not PO (but the manuscript says PO, p 3 line 132 and p 4 line 134, line 135). Which assay was used to assess the anti-Xa level of danaparoid? Was that assay calibrated against danaparoid? Most regular assays are not calibrated for this drug. Kitchen S et al, Thromb Haemost 1999;82:1289 have identified false low anti-Xa activity for Xa-clotting tests which they recommend to not use for assessing the danaparoid level.
  3. The authors state that they found autoantibodies against PF4 (p 6 line 239, p 7 line 281). The manuscript contains no evidence for this. I would suggest to omit “auto”. This is of no harm for the general content of the paper.

Minor

  1. COVID-19 should not be written as “Covid-19” (manuscript title and elsewhere in the paper).
  2. I assume all three patients became symptomatic after the first vaccination. This should be stated in the Abstract (line 27) and when presenting their history (p 2 line 78, p 3 line 115 and p4 line 145).
  3. The platelet count is n/µl, but given as “n/ml” throughout the text (e.g. p 2 line 85 and elsewhere).
  4. P 3, line 126, patient #2 had negative test results for thrombophilia including “GluP2000”. What does that mean? I never came across this marker.
  5. P 3, line 128. For the average reader of the paper it would be good to get the specific sinus as full wording rather than SSS, TS etc.
  6. Table 1, page 5. Please replace “Factor II” with “prothrombin G20210A” and please replace “Factor V” with “Factor V R5066 (Leiden)”. The wording “thrombocyte count” is very unusual, I would prefer “platelet count”.
  7. Page 7 has a paragraph on TTP (lines 273-277). Why?

Author Response

Reviewer #2

The authors are grateful for the proposals of reviewer #2. The manuscript has been modified as follows.

<< The authors should be very specific with the tests that were applied! Most of the relevant tests are very specific to the manufacturer and cannot be compared between laboratories. Which PF4-ELISA was used (manufacturer & modifications, if any); which SARS-CoV-2 antibody test (manufacturer & modifications, if any) was used; which antiphospholipid antibody testing (which antibodies where tested, beta2-gylcoprotein I and cardiolipin both as IgG and IgM I assume, in which assay), Did the authors also test for a functional lupus anticoagulant? In addition, please make clear how protein S was assessed (free protein S would be the recommended test, but that doesn’t give you “protein S acitivity”), same for protein C (chromogenic assay would be the recommended test). Please state if patients were tested for antithrombin activity. It was maybe low, which can be expected in the clinical setting. Apparently some type of HIPA was performed. Please cite a reference for this test.

>> The following laboratory tests had been used:

Anti-factor Xa (anti-Xa) assay: COAMATIC HEPARIN (Chromogenix) calibrated for low-molecular-weight heparin (LMWH) or rivaroxaban or apixaban, a calibration for danaparoid is not available. The anti-factor Xa-activity calibrated against LMWH was used, which might yield false-low results.

Antithrombin activity: Berichrom Antithrombin III (A) (Siemens)

ARCHITECT SARS-CoV-2 IgG (Abbott)

Beta-2-glycoprotein I antibody test: ELISA (Euroimmun)

Cardiolipin antibody: ELISA (Euroimmun)

Hemostasis analyzer: CS5100-Gerät (Sysmex)

HIPA, according to Greinacher et al. (https://www2.medizin.unigreifswald.de/ transfus/fileadmin/user_upload/doku_thrombo_gerinnung/anleitung_hipa.pdf)

Lupus anticoagulant (functional): Pathromtin  SL (aPTT) (Siemens) and LA 1 Screen (dRVVT) / LA2 Confirm (dRVVT) (Siemens)

PF4-IgG EIA (Immucor)

Protein C activity: ProC AcR (Siemens)

Protein S activity: Protein S Ac (Siemens)

SARS-CoV-2 antibody test:  LIAISON SARS-CoV-2 S1/S2 IgG (DiaSorin)

<< The authors explain that with danaparoid, they did not reach sufficient factor Xa inhibition. I assume that danaparoid was given SC and not PO (but the manuscript says PO, p 3 line 132 and p 4 line 134, line 135). Which assay was used to assess the anti-Xa level of danaparoid? Was that assay calibrated against danaparoid? Most regular assays are not calibrated for this drug. Kitchen S et al, Thromb Haemost 1999;82:1289 have identified false low anti-Xa activity for Xa-clotting tests which they recommend to not use for assessing the danaparoid level.

>> The following has been changed and added: Since a HIT was suspected, the medication was changed to 3x 750 mg danaparoid SC daily, which did not achieve a sufficient factor Xa inhibition, as measured by a non-calibrated anti-factor Xa assay. The same was true for the increased dosage of 4x 1250 mg danaparoid SC daily. The non-calibrated anti-factor Xa assay may yield false-low results.

<< The authors state that they found autoantibodies against PF4 (p 6 line 239, p 7 line 281). The manuscript contains no evidence for this. I would suggest to omit “auto”. This is of no harm for the general content of the paper.

>> The term “autoantibody/autoantibodies against PF4” was replaced by “antibody/antibodies”.

<<COVID-19 should not be written as “Covid-19” (manuscript title and elsewhere in the paper).

>> “Covid-19” was replaced by “COVID-19” in the entire manuscript

<< I assume all three patients became symptomatic after the first vaccination. This should be stated in the Abstract (line 27) and when presenting their history (p 2 line 78, p 3 line 115 and p4 line 145).

>> The word “first” was added whenever reasonable

<< The platelet count is n/µl, but given as “n/ml” throughout the text (e.g. p 2 line 85 and elsewhere).

>> This is a formatting mistake. It most likely occurred during the transformation from the submitted manuscript to the template. In the submission platelet counts are given as “µ/L” throughout.

<< P 3, line 126, patient #2 had negative test results for thrombophilia including “GluP2000”. What does that mean? I never came across this marker.

>> “GluP2000” was deleted.

<< P 3, line 128. For the average reader of the paper it would be good to get the specific sinus as full wording rather than SSS, TS etc.

>> Abbreviations of the sinuses were replaced by their full names in the entire manuscript.

<< Table 1, page 5. Please replace “Factor II” with “prothrombin G20210A” and please replace “Factor V” with “Factor V R5066 (Leiden)”. The wording “thrombocyte count” is very unusual, I would prefer “platelet count”.

>> The concerning changes can be found in table 1

<< Page 7 has a paragraph on TTP (lines 273-277). Why?

>> In the submission TTP comes without a paragraph

Reviewer 3 Report

To the authors:

In this paper, Wolf et al present very interesting data cases with cerebral venous thrombosis after vaccination with AstraZeneca. The clinical manifestation of these cases are well described. Data on the management of the patients are very important. In general, this paper is well written and contains novel aspects. This Reviewer has only some minor suggestions.

  1. Please provide data on fibrinogen levels, if available.
  2. More data on the used assays should be provided such as ELISA for the detection of PF4-heparin antibodies.
  3. It would also interesting if the authors could check the heparin dependency of the ELISA results by test antibody binding in the presence of high heparin concentration.
  4. The authors treated VIPIT patients with heparin. This observation is very important and should be clearly reported in the abstract.
  5. The dynamic of the platelet counts and D-Dimer after heparin therapy should be described. Did these parameter improved?

Author Response

Reviewer #3

<< Please provide data on fibrinogen levels, if available.

>> Fibrinogen levels have, unfortunately, not been measured

<< More data on the used assays should be provided such as ELISA for the detection of PF4-heparin antibodies.

>> The following has been added: PF4-IgG EIA (Immucor);

<< It would also interesting if the authors could check the heparin dependency of the ELISA results by test antibody binding in the presence of high heparin concentration.

>> The following has been added: PF4-IgG EIA (Immucor) (in the presence of high heparin concentration the ELISA was negative)

<<The authors treated VIPIT patients with heparin. This observation is very important and should be clearly reported in the abstract.

>> The following was added: Under treatment with low-molecular-weight heparin platelet counts normalized within several days.

<<The dynamic of the platelet counts and D-Dimer after heparin therapy should be described. Did these parameters improved?

>>The following was added: All patients received low-molecular-weight heparin (LWMH). They showed a continuous improvement of the platelet count and thrombocytopenia spontaneously resolved within < 2 weeks during the following course without immunoglobulin administration or plasmapheresis.